# Cardiac Hypertrophy and Related Dysfunctions in Cushing Syndrome Patients—Literature Review

**DOI:** 10.3390/jcm11237035

**Published:** 2022-11-28

**Authors:** Akinori Kanzaki, Manabu Kadoya, Satoru Katayama, Hidenori Koyama

**Affiliations:** 1Department of Internal Medicine, Hyogo College of Medicine, Sasayama Medical Center, Sasayama 669-2321, Hyogo, Japan; 2Department of Diabetes, Endocrinology and Clinical Immunology, Hyogo College of Medicine, Nishinomiya 663-8131, Hyogo, Japan

**Keywords:** adrenal tumor, autonomous cortisol secretion, cardiac functional disorder, cardiac hypertrophy, cardiomyocytes, Cushing syndrome, glucocorticoid excess, surgery, mortality

## Abstract

The survival rate of adrenal Cushing syndrome patients has been greatly increased because of the availability of appropriate surgical and pharmacological treatments. Nevertheless, increased possibility of a heart attack induced by a cardiovascular event remains a major risk factor for the survival of affected patients. In experimental studies, hypercortisolemia has been found to cause cardiomyocyte hypertrophy via glucocorticoid receptor activation, including the possibility of cross talk among several hypertrophy signals related to cardiomyocytes and tissue-dependent regulation of 11β-hydroxysteroid dehydrogenase type 1. However, the factors are more complex in clinical cases, as both geometric and functional impairments leading to heart failure have been revealed, and their associations with a wide range of factors such as hypertension are crucial. In addition, knowledge regarding such alterations in autonomous cortisol secretion, which has a high risk of leading to heart attack as well as overt Cushing syndrome, is quite limited. When considering the effects of treatment, partial improvement of structural alterations is expected, while functional disorders are controversial. Therefore, whether the normalization of excess cortisol attenuates the risk related to cardiac hypertrophy has yet to be fully elucidated.

## 1. Introduction

Since Harvey Cushing initially reported a 23-year-old woman with a pituitary adenoma in 1910, 50% of affected patients have died within five years of diagnosis, mainly because of a heart attack-inducing cardiovascular event, as a normal history of Cushing syndrome (CS) [1]. Numerous investigations regarding early diagnosis and appropriate treatment have been performed, with some small-population studies showing relative long-term mortality (within five to seven years) after treatment in patients with CS similar to that in the general population [2,3,4,5]. Nevertheless, diagnostic delay remains a serious issue [6] and a number of studies have found that the survival rate of CS patients has been compromised after successful treatment [3,7,8,9,10,11,12]. In particular, a large-scale population study revealed a higher frequency of life-threatening events such as heart attack, with a standardized mortality ratio of 3.0 (95% CI: 2.4–3.7) among 1127 patients with benign adrenal CS [13].

Since the 1990s, subclinical adrenal CS, which features autonomous secretion of excess cortisol without detectable related physical characteristics [14], has gained attention. In 2016, a European Society of Endocrinology Clinical Practice Guideline was presented that endorses this pathophysiology as “autonomous cortisol secretion (ACS)” and “possible (p)ACS” [15]. Those conditions are on the same spectrum and classified based on the results of a 1-mg dexamethasone test (DST), with ACS showing >138 nmol/L (>5.0 µg/dL) and pACS 51–138 nmol/L (1.9–5.0 µg/dL). These pathological conditions have also been shown to have a relationship with an increased risk of diabetes mellitus [16], hypertension [17], and heart attack-inducing cardiovascular events [18,19,20], as well as overt CS, whereas pathophysiological differences have also been considered. For example, approximately two-thirds of cases of ACS do not develop into overt CS [21] and inherent differences in genetic background have been revealed, such as the somatic mutation of PRKACA that encodes the catalytic subunit α of cyclic adenosine monophosphate (cAMP)-dependent protein kinase (PKA), which is common in overt adrenal CS (35~66%) cases, though uncommon in ACS (~11%) [22,23].

Cardiac hypertrophy is a well-known condition that indicates the possibility of hypercortisolemia-inducing ACS. Because of its relationship with induction of heart failure and cardiac death, timely detection of such an alteration can lead to early diagnosis and contribute to reducing mortality. However, there are few literature reviews regarding related experimental mechanisms or the clinical features of hypercortisolemia. The aim of the present study was to provide an updated review of findings obtained in both experimental and clinical medicine investigations showing the influence of excess cortisol on heart structure and function, as well as to summarize the effects of therapeutic interventions on cardiac impairment in affected patients.

## 2. Physiology of Cortisol

### 2.1. Glucocorticoid Receptor and Regulating Enzymes

The effects of bioactive cortisol, which is synthesized in the adrenal cortex, are exerted following connection to a glucocorticoid (GC) receptor (GR) or mineralocorticoid receptor (MR). GR belongs to a ligand-dependent transcription factor nuclear receptor superfamily that is comprised of three functional domains: an N-terminal transactivation domain (NTD), central DNA-binding domain (DBD), and C-terminal ligand-binding domain (LBD) [24]. Nearly all human tissues and organs express GR, which forms a multiprotein complex in cytoplasm [25]. Cortisol binding induces GR nuclear translocation with dissociation of the complex, including heat shock protein 90 [26], after which GC signaling enhances genomic and rapid non-genomic effects. MR is also a member of the ligand-dependent transcription factor nuclear receptor superfamily, showing a structural homology with GR, and induces genomic and rapid non-genomic effects in the same way as GR, though the outcome differs largely from that of GR activation and limited localization has been shown, such as cortical collecting ducts, epithelium of the large intestine and cerebrum, endothelium of blood vessels, and cardiac muscle tissue. Although both cortisol and aldosterone can function as a ligand to MR, it should be noted that cortisol has 100- to 1000-fold higher concentration in blood, and a 10- to 30-fold higher affinity to MR as compared with aldosterone. Furthermore, cortisol potentially shows both agonist and antagonist characteristics for MR [27]. However, MR transcription is seldom activated without stress or tissue injury [28,29,30], and GC was shown to function as an MR agonist in a rat heart failure model [31].

To produce an appropriate effect of cortisol and aldosterone, 11β-hydroxysteroid dehydrogenase type 1 and 2 (11β-HSD1, 2) play essential roles [32,33]. 11β-HSD1 converts inert 11-keto forms (cortisone, 11-dehydrocorticosterone) into cortisol by using the co-substrate NADPH provided by hexose-6-phophate dehydrogenase (H6PDH) [34], which results in an increase in local active cortisol concentration [35,36,37]. On the other hand, 11β-HSD2 inactivates cortisol [38,39], and has a high affinity but low capacity for NAD-dependent dehydration [40,41]. Therefore, the degree of the effect of cortisol on individual target cells is dependent on the balance of an intracellular GC-activating enzyme (11β-HSD1) and -inactivating enzyme (11β-HSD2) [42], with 11β-HSD1 inferior to 11β-HSD2 in regard to the strength of cortisol binding. For example, a suppressed effect of GC would be expected in the kidneys, where 11β-HSD2 is significantly expressed, unlike in the heart [28].

### 2.2. Physiological Role of Glucocorticoid in Rodent Cardiomyocytes

Endogenous GC contributes to maintain heart performance for regulating the life cycle of cardiomyocytes involved in growth, differentiation, metabolism, and apoptosis [43]. GC action via GR matures fetal cardiomyocytes and myofibrillar, and boosts its mitochondrial activation, by which contractile force is reinforced [44]. Narayanan et al. assessed the potential therapeutic benefits of dexamethasone treatment on myocardial function in senescent rats and demonstrated that it can reverse contractile performance by approximately two-fold caused by increased uptake of ATP-energized Ca2+ in the sarcoplasmic reticulum [45]. In addition, GC potentially inhibits cardiomyocyte apoptosis [46,47] by activating serum and glucocorticoid-responsive kinase (SGK-1), B-cell lymphoma-extra large (Bcl-xL), and growth arrest specific 2 (Gas2) [48,49]. Additionally, GR potentially protects cardiomyocytes from DNA damage and drug-induced cell death by regulating the expression of Kruppel-like factor 13 (KLF13), a major mediator of GR [50].

H6PDH activity has been identified in cardiomyocytes and fibroblasts in rats [51,52], while HSD11B1 mRNA has been observed in human hearts [53], though a physiologically normal condition restricts GC regeneration [54,55,56]. Nevertheless, 11β-HSD1 is essential for maintaining contractile force generation [54] and heart growth [57]. Interestingly, White et al. reported that cardiomyocytes lacking 11β-HSD1 in perinatal mice matured with a shortened length, though a phenotypically normal heart was successfully developed in those animals [58]. In relation to these findings, Rahman et al. presented the possibility of a reduction in left ventricular mass caused by a single nucleotide polymorphism in the HSD11B1 gene [59].

### 2.3. Harmful Effects to Cardiomyocytes by Experimental Excessive Glucocorticoid Exposure

#### 2.3.1. Excessive Glucocorticoid and Cardiomyocyte Hypertrophy

Regarding the effects of excessive GC in therapeutic and pathologic conditions, abundant knowledge showing its influence with exposure to the heart has been accumulated. For example, antenatal corticosteroid therapy, known to reduce neonatal death [60], has been found to have a limited contribution to preterm birth [61], whereas it is associated with high risks of cardiovascular disease, hypertension, and type 2 diabetes in adults [62,63,64,65,66,67]. Ren et al. explored the direct effects of excessive GC on cardiomyocytes by exposing a rat embryonic cardiomyocyte cell line (H9C2) and primary cardiomyocytes to dexamethasone [50]. The results showed a significant increase in cell size along with elevated levels of expression of cardiomyocyte hypertrophic markers, such as atrial natriuretic factor (ANF), β-myosin heavy chain (β-MHC), and skeletal muscle α-actin (α-SKA) (Figure 1). Moreover, Ingenuity Pathway Analysis revealed that 58 genes were associated with cardiomyocyte hypertrophy signaling at 48 h after dexamethasone treatment. Additionally, dexamethasone was found to exhibit an anti-apoptotic effect on cardiomyocytes with exposure to tumor necrosis factor (TNF)-α with serum deprivation, while its deprivation abolished the effect of dexamethasone to elevate the expression levels of those hypertrophic maker genes, except for β-MHC. These results were diminished by addition of a GR antagonist or knock-down of GR expression, while suppression of MR activity did not have such effects, indicating the essential relationship of GR activity. Lister et al. presented findings that verified altered corticosteroid signaling in cardiomyocyte hypertrophy induced by phenylephrine (α-adrenergic receptor agonist) and phorbol ester (protein kinase C (PKC) activator), and as such signal activation is known to be involved in hypertension and diabetes [68]. Consequently, a hypertrophic response was found to accompany a significant increase in atrial natriuretic peptide mRNA (8- to 12-fold increase in both) and rDNA transcription (2-fold increase), which exhibit corticosteroid effects, and also GR and MR expression (2-fold increase) with enhanced receptor signaling. When priming with phenylephrine was performed, corticosteroids potentiated a hypertrophic response via GR, while phorbol ester-induced hypertrophy exhibited increased 11β-HSD1 expression and its reductase activity. These results indicate cross talk between corticosteroid receptor-activated pathways, and both α-adrenergic and PKC signals (Figure 1).

On the other hand, intracellular sodium kinetics in ischemia, heart failure, and hypertrophy has been considered [69,70,71,72,73]. According to a report by Katoh et al., cultured neonatal rat ventricular myocytes were treated for 24 h with corticosterone, aldosterone, and dexamethasone, and the results showed a nearly 1.5-fold increase in intracellular sodium concentration, though that occurred in a dose-dependent manner [74]. In addition, a GR antagonist reduced intracellular sodium concentration, and positive correlations between hypertrophic gene expressions and concentration were observed, findings not obtained with an MR antagonist. Additional experiments then revealed that dexamethasone upregulated the mRNA of sodium-calcium exchanger (NCX)1 and its protein, which is related to intracellular sodium homeostasis and influences calcium efflux in cardiomyocytes [72]. These findings suggest that exposure to excessive GC increases the concentration of intracellular sodium via GR, while dexamethasone treatment explains, at least in part, the direct effect of that increase, leading to cardiomyocyte hypertrophy (Figure 1).

#### 2.3.2. Tissue-Specific Role of 11β-Hydroxysteroid Dehydrogenase Type 1 in Glucocorticoid Excess

When considering local regulation of GC, its association with 11β-HSD1 is another related issue. Nishiyama et al. recently confirmed the influence of 11β-HSD1 in mouse tissues showing persistent GC excess [75]. Two weeks of administration of excessive corticosterone to male mice decreased expression levels of 11β-HSD1 mRNA in the hippocampus and liver, while those were increased in abdominal adipose tissue. Furthermore, similar results were obtained with male corticotropin releasing hormone (CRH)-overexpressing transgenic mice, an animal model of CD [76,77], while performance of an adrenalectomy reversed those changes, suggesting a tissue-dependent action (Figure 1). Huang et al. also noted the role of 11β-HSD1 in cardiomyocytes [78]. Briefly, treatment of primary neonatal rat ventricular cardiomyocytes (NRCMs), which show upregulated 11β-HSD1 expression, with palmitic acid induced a significant enlargement of cell size and increased the mRNA levels of cardiomyocyte hypertrophy-specific genes, including ANF, SKA, and β-MHC, whereas either a selective inhibitor of 11β-HSD1-treated or 11β-HSD1-deficient NRCMs caused a decrease in cell size. Moreover, they also confirmed marked attenuation of 11β-HSD1-induced hypertrophy of cardiomyocytes in not only the presence of a GR antagonist (RU486), but also with the MR antagonist spironolactone, which inhibits nuclear MR translocation [79].

## 3. Effects of Hypercortisolemia on Heart Structure and Function in Clinical Findings

In clinical cases, the pathophysiological relationship of hypercortisolemia with cardiac hypertrophy is complex and studies that assessed the structural alterations of cardiac muscle including hypertrophy in CS patients have been presented. Although patients with adrenocorticotrophic hormone (ACTH)-dependent and -independent CS participated in those investigations, few studies have focused on overt adrenal CS, while knowledge regarding cardiac hypertrophy in ACS cases is also quite limited (Table 1, Figure 2).

### 3.1. Heart Structural Alterations in Patients with Overt Cushing Syndrome

A common finding in echocardiographic studies of patients with overt CS is left ventricular hypertrophy (LVH), known as a risk factor for myocardial ischemia and development of heart failure [88]. Although the population was small, Sugihara et al. examined 12 patients (27–57 years old; eight females, four males) with overt CS, including nine with adrenal CS, and provided suggestive results [80]. Nine of those patients showed LVH with an interventricular septum thickness (IVST) diameter of >15 mm at end-diastole (mean ± standard deviation, 17.8 ± 6.2 mm), which was more severe than that in patients with essential hypertension (*n* = 53, 11.5 ± 2.9 mm, *p* < 0.01) or primary aldosteronism (*n* = 8, 11.3 ± 1.2 mm, *p* < 0.01). Interestingly, posterior wall thickness (PWT) diameter at end-diastole was relatively mild in all groups (11.3 ± 1.8 mm in CS group, 10.4 ± 1.6 mm in essential hypertension group, 10.8 ± 1.5 mm in primary aldosteronism group). Furthermore, a significant higher ratio of IVST to PWT (IVST/PWT) exceeding 1.3 was seen only in the CS group, suggesting existing asymmetric septal hypertrophy (ASH). Such disproportional hypertrophy is uncommon in both essential hypertension [89,90] and secondary hypertensive disease cases, including renovascular hypertension [91,92], primary aldosteronism [92], and pheochromocytoma [93]. A possible mechanism for this structural alteration has been speculated to be related to the increased rate of regional stress [94], while loss of diurnal pattern of cortisol causing nighttime high blood pressure (BP) [95] might also have an effect on this change. However, that was a small-scale study and at least two other studies have reported contrasting evidence. First, Muiesan et al. surveyed 42 patients with CS (36 with CD, 6 with adrenal adenoma, 1 each with adrenal carcinoma and ectopic ACTH production) and noted significantly larger LV interventricular septum diameter (IVSd) (10.1 ± 2.2 vs. 8.8 ± 1.3 mm, *p* = 0.05) and LV posterior wall diameter (PWd) (9.8 ± 1.7 vs. 7.8 ± 1.1 mm, *p* = 0.005) measurements as compared with control subjects matched for age, sex, body mass index (BMI), smoking habit, lipid level, BP level, and duration of hypertension [81]. In another study, Toja et al. noted that both IVST (10.2 ± 0.17 vs. 9.4 ± 0.17 mm, *p* < 0.05) and PWT (9.7 ± 0.16 vs. 8.5 ± 0.12 mm, *p* < 0.05) were significantly thicker in patients with CS than controls matched for age, sex, and BMI, while none of the CS patients presented ASH [82]. The results of those studies suggest that asymmetric change is not an essential phenotype in CS cases.

Although LVH is a well-known finding in CS patients, recent studies have revealed the frequency to be less than 50% [81,82,83], while relative wall thickness (RWT) and concentric remodeling, an LV geometric pattern category showing normal LV mass (LVM) but with concentric geometry [96], have recently received attention. A ten-year follow-up study that evaluated the relationships of LVM and concentric geometry with cardiovascular events and death in patients with essential hypertension found a 2.3-fold greater risk of cardiovascular events per 100 patient-years, even in cases with a concentric remodeling state, as compared with those with normal geometry [97]. Furthermore, 3% of the patients in that study with concentric remodeling died due to progression of hypertrophy. Therefore, RWT and concentric remodeling may also be of great concern for CS patients. Toja et al. divided CS patients and a control group into hypertension and normotension sub-groups [82]. Their results showed that RWT was indeed greater in both hypertensive (0.44 ± 0.01 vs. 0.36 ± 0.01) and normotensive (0.41 ± 0.01 vs. 0.36 ± 0.02) CS patients as compared with the controls (both *p* < 0.05). Additionally, among the CS patients, the hypertensive group had a significant greater score as compared with the normotensive group. However, LV mass index (LVMI, generally described as LVM/body surface area) in the CS with normotension group was similar to that seen in the control group (42.9 ± 2.34 vs. 39.3 ± 1.91 g/m^2.7^), while that was significantly larger in the CS with hypertension group (49.3 ± 2.12 mm) as compared with the normotensive CS (*p* < 0.05) and control with hypertension (42.1 ± 1.61 mm, *p* < 0.05) groups. In another study, Kamenický et al. demonstrated that LVMI is associated with systolic BP (sBP) [120 (113; 130) mmHg (median and interquartile range) in CS group vs. 112 (108; 117) mmHg in control group, *p* = 0.009; r = 0.72, *p* = 0.045] [84]. Indeed, hypertension can cause pressure overload leading to heart failure through LV remodeling rather than attenuation of myocardial function [98], while arterial hypercoagulability seen in CS [99] potentially exacerbates hypertension. On the other hand, Fallo et al. reported findings showing that RWT was not related to BP level in patients with CS [100]. Additionally, Avenetti et al. found that neither LVM nor geometry had an association with 24 h ambulatory BP (ABP) monitoring, with no difference in all-day BP values between CS and control groups (CS group: 138 ± 14/85 ± 10 mmHg, control group: 134 ± 15/84 ± 11 mmHg; *p* = 0.36 and 0.66, respectively), though a significant greater prevalence of pressure non-dipping profile in those CS patients was noted (56% vs. 16%, *p* < 0.05) [85]. Together, these findings indicate that the effects of a hypertensive condition remain to be fully revealed and might depend, at least in part, on duration or severity.

### 3.2. Cardiac Dysfunction in Patients with Overt Cushing Syndrome

In addition to structural alterations, cardiac dysfunction in CS patients is another factor for consideration. Toja et al. found no significant difference for left ventricular ejection fraction (EFLV), or ratio of early (E) and late (A) wave diastolic filling velocity between CS cases and a control group with normotension (65.2 ± 0.65% vs. 65.9 ± 0.42%, 1.37 ± 0.10 vs. 1.48 ± 0.06, respectively), while a significant attenuation of EFLV was only observed in the CS group with hypertension as compared with the controls with hypertension (63.7 ± 0.51% vs. 67.4 ± 0.74%, *p* < 0.05) [82]. In addition, midwall fractional shortening (mwFS), an index of absolute and stress-adjusted systolic dysfunction, was significantly attenuated in 20% of the CS patients as compared with the control group regardless of BP level (normotensive: 17.4 ± 0.41% vs. 18.5 ± 0.52%, *p* < 0.05; hypertensive: 16.4 ± 0.28% vs. 18.8 ± 0.47%, *p* < 0.05) [82]. Muiesan et al. also reported that mwFS in CS patients was significantly decreased as compared with BP level, duration of hypertension, and other similar factors in a matched control group (16.2 ± 3.0% vs. 20.5 ± 4.5%, *p* < 0.01), as was stress-adjusted endocardial FS [81]. Additionally, they found that mwFS was independently associated with CS (F = 22.6, *p* < 0.001) and hypertension (F = 5.78, *p* = 0.012), while no significant difference was observed in regard to endocardial FS. In a comparison of LV diastolic function between CS patients and controls, a reduced ratio of E/A velocity (0.94 ± 0.27 vs. 1.18 ± 0.32, *p* = 0.03), along with decreased E velocity (55.6 ± 12 cm/s vs. 78 ± 21 cm/s, *p* = 0.005) and prolonged E-wave deceleration time (205 ± 36 ms vs. 150 ± 30 ms, *p* < 0.001) were observed. Notably, analysis regarding the association of those diastolic parameters to CS and hypertension revealed that E-wave deceleration time was independently associated with CS (F = 12.9, *p* = 0.001) but not with hypertension (F = 1.93, *p* = 0.18), while E/A ratio was independently associated with both CS (F = 12.9, *p* = 0.001) and hypertension (F = 17.3, *p* < 0.001). Finally, the authors concluded that the cardiac structural alterations described above are associated with attenuation of both systolic and diastolic performance, such as mwFS, E/A ratio, and E-wave deceleration time, which may partially explain the high risk of heart attack occurring in patients with CS.

### 3.3. Cardiac Magnetic Resonance Imaging Test for Evaluation of Patients with Overt Cushing Syndrome

Cardiac magnetic resonance (CMR) imaging testing has been shown to provide results that are useful for highly accurate and comprehensive evaluations of cardiac geometry and function, as well as myocardial fibrosis [101,102,103], and those cardiac parameters have been assessed in patients with CS [84]. Comparisons of LV and right ventricular (RV) parameters in healthy controls and CS patients matched for age, sex, and BMI showed similar results for both parameters, as end-systolic volume was higher (LV; 31.8 (24.2; 37.4) mL/m^2^ vs. 22.4 (16.7; 29.4) mL/m^2^, *p* = 0.002; RV; 36.6 (32.6; 42.5) mL/m^2^ vs. 27.1 (21.6; 35.3) mL/m^2^, *p* = 0.012), though end-diastolic volume was not significantly different. Therefore, stroke volume index was significantly smaller in the RV (34.1 (26.9; 40.3) mL/m^2^ vs. 41.9 (37.1; 48.3) mL/m^2^, *p* = 0.033) and tended to be smaller in the LV (34.9 (31.4; 46.4) mL/m^2^ vs. 44.8 (37.5; 49.7) mL/m^2^, *p* = 0.079), and lower EF values were also observed (EFLV: 55.0 (51.0; 58.4)% vs. 64.3 (60.9; 69.6)%, *p* < 0.01; EFRV: 49.3 (42.5; 53.4)% vs. 60.0 (54.6; 64.3)%, *p* < 0.01). Additionally, left atrial (LA) parameters showed a significant increase in minimal volume (20.8 (15.5; 247)) mL/m^2^ vs. 15.6 (13.1; 20.7) mL/m^2^, *p* = 0.042) but not in maximal volume; consequently, EFLA was markedly lower in the CS patients as compared with the controls (39.2 (34.8; 48.4)% vs. 56.1 (48.6; 61.2)%, *p* < 0.01). Finally, end-diastolic LV segmental thickness shown in basal, midventricular, and apical slices was markedly increased (11.2 (8.96; 12.2) mm vs. 7.99 (6.78; 8.61) mm, *p* < 0.01; 10.3 (8.43; 11.3) mm vs. 7.26 (6.33; 8.21) mm, *p* < 0.01; 8.45 (7.86; 9.52) mm vs. 5.90 (5.16; 6.68) mm, *p* < 0.01, respectively). Furthermore, late gadolinium enhancement testing, which depicts dense myocardial fibrosis [101], revealed no delayed enhancement, suggesting the absence of dense replacement myocardial fibrosis in the CS cases. Nevertheless, the authors noted a limitation related to the method, in that the presence of diffuse interstitial fibrosis could not be precluded. Altogether, this CMR study reinforced evidence regarding LV structure and function as well as echocardiographic results, and further provided informative findings related to RA and LA systolic functions.

### 3.4. Contributions of Human Serum and Urinary Cortisol Levels to Cardiac Impairment

It is rather surprising that the associations of serum and urinary cortisol levels with cardiac hypertrophy in humans have yet to be clarified. A multivariate analysis of patients with adrenal CS (*n* = 50) conducted by Takagi et al. demonstrated that serum cortisol (24.3 ± 16.6 µg/dL) and potassium (3.8 ± 0.6 mEq/L) levels independently contributed to LVMI (r2 = 0.647; *p* < 0.05 and *p* < 0.01, respectively), while hemoglobin A 1c (HbA1c) (6.2 ± 1.6%) and serum potassium levels independently contributed to EFLV (r^2^ = 0.529; *p* < 0.05 and *p* < 0.05, respectively) [97]. They also explained that the diverse mechanisms of cortisol had indirect effects on such factors as hypertension [104], noradrenalin and angiotensin II [105], and the local RAA system [106], as well as direct effects, such as stimulation of cell cycle activity in cardiomyocytes leading to proliferation [107] and accelerating of cardiomyocyte hypertrophy by expression of angiotensinogen mRNA [108]. Moreover, Toja et al. presented multifactorial analysis of variance results, along with a theory that cortisol excess and hypertension interact to worsen cardiac parameters (e.g., F = 7.7, *p* < 0.05 for RWT) [82]. The authors considered this to be due to the 3.01-fold higher risk of increase in LVMI seen in CS patients with hypertension as compared with hypertensive controls, and because patients with cured CS with residual hypertension still showed significant cardiac impairments in both structural and functional parameters (LVMI, IVSd, PWd, RWT, EFLV, mwFS, E/A) as compared with the hypertensive controls. However, a separate series of studies revealed no correlation of cortisol level with impairment in patients with CS, based on the following findings: (1) the levels of urinary 17-hydroxycorticosteroids, serum cortisol, urinary adrenalin and noradrenalin, plasma aldosterone, and plasma renin activity were not correlated with IVST or PWT [80]; (2) there was no correlation of urinary cortisol with RWT, LVMI, or diastolic disfunction [81]; (3) 24 h urinary cortisol (1016.72 ± 549 nmol/24 h) was not associated with LVM (r = 0.1, *p* = 0.5), RWT (r = 0.02, *p* = 0.89), or all-day BP values [100]; and (4) CMR test results of LV and RV function also had no correlation with biological variables, including 24 h urinary cortisol and plasma cortisol levels [84]. Presently, the underlying mechanisms related to cortisol level alterations remain controversial and complex, and further study is needed, including an analysis matched for duration of CS as well as hypertension.

### 3.5. Geometric and Functional Impairment in Autonomous Cortisol Secretion Patients

There is only limited knowledge regarding alterations of heart structure and function associated with ACS, though its association with a high risk of heart attack-inducing cardiovascular events has been noted. Iacobellis et al. examined 46 consecutive patients (22 males, 24 females) with incidentaloma, including 40 with non-functional incidentaloma (mean age 61.6 ± 10.7 years) and six with mild adrenal CS (mean age 69.3 ± 12.5 years), defined when two or more of the following abnormalities were present: urinary-free cortisol level > 70 µg/day, >5 µg/dL of serum cortisol after 1 mg DST, and ACTH level < 10 pg/mL in the morning (08:00) [86]. The study protocol excluded patients with known malignancy or hormone-secreting tumors, such as pheochromocytoma and/or aldosterone-secreting adenoma, as well as severe or paroxysmal hypertension, hypokalemia, or clinical signs of hypercortisolism or hyperandrogenism. Echocardiographic examinations were performed to determine LVM and epicardial fat thickness, the latter of which reflects visceral fat deposition in the heart [109,110] and thought to be an indicator of cardiovascular risk [111,112,113], with the results compared with those of 30 healthy controls (mean age 59.6 ± 10 years, 20 males, 10 females). A significantly greater epicardial fat thickness and increase in LVM including height^2.7^-adjusted LVM, which is more sensitive for detecting LVH in patients with obesity [114], were observed (7.9 ± 0.8 mm vs. 7.4 ± 0.6 mm, *p* = 0.01; 49.7 ± 10 g/m^2.7^ vs. 47 ± 8 g/m^2.7^, *p* = 0.05; respectively). The results suggested that epicardial fat shown in echocardiogram findings was a non-invasive marker of earlier cardiac abnormalities in patients with adrenal incidentaloma. Interestingly, LVM^h2.7^ and LVM were obviously greater in mild CS patients than in patients with non-functional adenomas (62.6 ± 10 g/m^2.7^ vs. 47.7 ± 9 g/m^2.7^, *p* = 0.01; 223.7 ± 35 g vs. 200.5 ± 45 g, *p* = 0.01; respectively), whereas epicardial fat thickness was not significantly different (8.0 ± 0.7 mm vs. 7.9 ± 0.9 mm). On the other hand, Evran et al. found no echocardiographic differences in five patients (mean age 49.60 ± 8.64 years) with adrenal ‘subclinical CS (SCS)’, defined based on non-suppressive cortisol level (>50 nmol/L) in post-DST results, and ACTH < 10 pg/mL as compared to 76 patients (mean age 52.09 ± 9.78 years) with nonfunctional adrenal incidentaloma (NFAI) [115]. None of the 81 patients had a history of type 2 diabetes, hypertension, artery disease, dyslipidemia, drug abuse, steroid use, or functional adrenal mass (CS, primary hyperaldosteronism, pheochromocytoma). As compared with an age-, sex-, and BMI-matched healthy control group (*n* = 33), both the SCS and NFAI groups showed larger waist circumference (*p* = 0.00) and impaired glucose metabolism profiles (fasting plasma glucose and insulin levels, HOMA-IR; *p* = 0.01, 0.00, and 0.00, respectively), and higher triglyceride level (*p* = 0.01). The differences in hormonal parameters between the patients with SCS and NFAI reflected pathology, with no differences for basal serum level of cortisol, ACTH, or dehydroepiandrosterone sulfate (DHEAS), but a higher post-DST cortisol level in the SCS group (4.84 ± 0.47 vs. 1.32 ± 0.09, *p* = 0.00). When assessing echocardiographic measurements between the SCS and NFAI groups, there were no differences found for IVST (11.20 ± 2.49 mm vs. 10.38 ± 1.47 mm), PWT (10.80 ± 1.78 mm vs. 10.31 ± 1.45 mm), or end-diastolic diameter (49.00 ± 2.00 mm vs. 47.5 ± 3.49 mm).

Of note, a recent study by Sbaedella et al. disclosed a mild correlation between serum cortisol level especially in post-DST and LVH in echocardiography in patients with pACS [87]. Seventy-one consecutive patients with adrenal incidentaloma were enrolled in that study who were not affected by an overly active or malignant adrenal tumor, including overt CS, pheochromocytoma, Conn syndrome, adrenal carcinoma, late-onset congenital adrenal hyperplasia, myelolipoma, adrenal metastasis, and adrenal hemorrhage cases. Furthermore, none had a history of myocardial infarction or unstable angina, psychiatric disease or alcohol abuse, or were taking medications affecting glucocorticoid production. To diagnose ACS, 1 mg-DST based on the 2016 European Society of Endocrinology guidelines [15] and a two-day 2 mg Liddle test (2 mg-DST), which has greater specificity than 1 mg-DST (97–100% vs. 87.5%) [116], were performed. All 71 participants underwent 1 mg-DST, then those showing a cortisol level > 50 nmol/L (1.8 µg/dL) (*n* = 37) underwent 2 mg-DST. Only 3 patients showed suppressed cortisol (≤50 nmol/L) and 34 were within a range of 51–138 nmol/L, considered to be pACS; thus, the remaining 37 were categorized as having a non-functioning adenoma (NFA). Their clinical and biochemical characteristics at the baseline were similar, such as mean age 68 (57–73) years in the pACS group vs. 67 (60–72) years in the NFA group, male sex percentage 38.2% vs. 29.7%, and BMI 26.7 (22.6–29.3) vs. 25.9 (23.7–31.2), while other metabolic or electrocyte profiles and cardiovascular risk factors were also similar, except for adenoma diameter (27.4 (23.4–31.5) mm vs 20.2 (17.2–23.1) mm, *p* = 0.004). As for hormone parameters including baseline serum cortisol, ACTH, aldosterone/renin ratio, 17OH progesterone, DHEAS, urinary free cortisol, androstenedione, urine metanephrine, and lower levels of ACTH (18 (15–20) pg/mL vs. 25 (19–31) pg/mL, *p* = 0.028) and DHEAS (37.5 (18–54) µg/dL vs. 77.8 (45–189) µg/dL, *p* = 0.012) were observed in the pACS group. A covariance model with fixed age, BMI, sBP, LDL cholesterol, and HbA1c, revealed geometric and functional differences as follows. (1) IVST and PWT were significantly thicker in the pACS as compared with the NFA group (10.7 (95% CI: 10.2–11.2) mm vs. 9.9 (95% CI: 9.4–10.4) mm, *p* = 0.015; 9.1 (95% CI: 8.5–9.6) mm vs. 8.4 (95% CI: 7.9–9.0) mm, respectively; *p* = 0.047), while LV end-diastolic diameter (LVDd) and RWT were not. (2) There was a greater increase in LVMI for both LV mass^h2.7^ and body surface area (BSA)-adjusted LVM (LVMBSA) confirmed in the pACS group (46.1 (95% CI: 42.5–49.6) g/m^2^ vs. 36.2 (95% CI: 32.8–39.6) g/m^2^, *p* = 0.01; 95.3 (95% CI: 88.0–102.5) g/m^2^ vs. 76.6 (95% CI: 69.6–83.6) g/m^2^, *p* = 0.001, respectively). (3) LVEF was similar between the two groups, while tricuspid annular plane systolic excursion (TAPSE), which partially represents RV function, was slightly reduced in the pACS patients, though the mean value was not highly suggestive of so-called RV dysfunction (<17 mm) (20.2 (95% CI: 18.9–21.6) mm). (4) Significantly higher values for septal and mean (lateral and septal) E/e’ were found in the pACS group (10.3 (95% CI: 9.0–11.6) vs. 8.3 (95% CI: 7.2–9.5), *p* = 0.028; 8.85 (95% CI: 7.8–9.8) vs. 7.4 (95% CI: 6.6–8.3), respectively; *p* = 0.044), though the difference for E/A was not significant. Furthermore, a stepwise method showed the main contributor to LVMI to be post-DST cortisol level, which accounted for 13.7%, followed by sBP (9.9%) and HbA1c (8.1%). When considering these findings, the higher prevalence of diastolic function abnormality, especially in septal E/e’, found in the pACS patients is notable. Elevation of E/e’ value generally follows a re-increase in E wave velocity after reduction in both E and e’ velocities. In this study, lateral, septal, and mean e’ were not significantly different between the pACS and NFA cases, though there was a tendency of relaxation impairment shown by septal and mean e’ values observed in the pACS group (7.4 (95% CI: 6.4–8.2) cm/s vs. 8.5 (95% CI: 7.7–9.3) cm/s, *p* = 0.059; 8.3 (95% CI: 7.4–9.2) cm/s vs. 9.4 (95% CI: 8.6–10.2) cm/s, respectively; *p* = 0.067), suggesting that the pACS patients might have been above a certain level of diastolic dysfunction.

On the other hand, A-wave velocity has received focus as a predictor of arterial stiffness [117] and also reported to be a useful index for evaluating LV diastolic disfunction [118], though related results for patients with adrenal CS are limited. We are currently conducting the Hyogo Sleep Cardio-Autonomic Atherosclerosis (HSCAA) study, a single-center cohort investigation performed at the hospital of Hyogo College of Medicine in Japan, to examine cardiovascular risk factors in patients with metabolic and endocrine diseases, including adrenal CS [119]. According to preliminary results of echocardiographic examinations, increased A-wave velocity has been observed in age- and metabolic profile-matched patients with adrenal ACS (mean age 58.8 ± 14.7 years, *n* = 25) as compared with those with a non-functioning adrenal adenoma (56.4 ± 13.9 years, *n* = 101) (75.3 ± 19.3 cm/s vs. 67.3 ± 15.5 cm/s, *p* = 0.04), while no significant differences for either e’ velocity or E/e’ between the groups have been confirmed (6.8 ± 2.4 cm/s vs. 7.1 ± 1.6 cm/s, *p* = 0.25; 10.2 ± 3.4 vs. 9.1 ± 2.8, *p* = 0.10, respectively). These findings might be useful in regard to early detection of ACS, though accumulation of additional cases is needed.

## 4. Treatment of Adrenal Cushing Syndrome and Effects on Cardiac Impairment

An adrenalectomy is recommended as standard care for overt adrenal CS, especially in unilateral CS cases, only a few studies with small populations have demonstrated low to very low levels of evidence regarding surgery for ACS [15]. Surgical treatment for ACS may contribute to reduce BP level and improve some metabolic profiles, such as glucose and lipid metabolism, as compared with conservative management [120,121,122,123], though differences related to long-term outcome have not been elucidated.

Nevertheless, structure alterations of the human heart in CS have been reported to be partially reversible after remission. Sugihara et al. followed nine patients after surgery for periods from three months to six years three months [80]. Consequently, seven showed decreases in IVST and PWT, and hypertension persisted in only two. Kamenický et al. presented results of an MRI study of 17 patients, 13 of whom reached surgical remission including all with adrenal adenomas and four who received steroidogenesis inhibitors, including their re-assessments at six months after remission [84]. The results showed improvements in both LV geometry, such as basal, midventricular, and apical wall thickness, and function including EFLV and stroke volume index. In addition, RV and LA functions were partially improved. On the other hand, Toja et al. performed follow-up examinations of 22 patients for a longer term after surgery (median 28.8 months). The results showed that functional parameters, including EF, mwFS, E/A ratio, E-wave deceleration time, and isovolumetric relaxation time, did not have significant recovery, though amelioration but not full recovery of IVST, PWT, LVMI, and RWT was demonstrated in association with improved BP level, as well as glucose and triglyceride profiles [82]. Together, these results suggest that functional improvement remains controversial, though it may follow structural amelioration.

Since the severity of LVH is correlated with cardiovascular risk, normalization of excess cortisol may contribute to reduce mortality. Nevertheless, additional studies focused on long-term outcomes of cured adrenal CS patients, including differences between surgery and conservative treatment, such as steroidogenesis inhibitor administration, are needed, while results of investigations of adrenal ACS cases are also crucial.

## 5. Conclusions and Perspective

In summary, excess cortisol has been experimentally shown to induce cardiomyocyte hypertrophy via activated GC signaling through GR. On the other hand, the pathophysiology in humans is more complex, indicating associations with several different factors including hypertension. Based on this literature review, it is concluded that a hypertensive state may exacerbate LVH in CS patients, though is possibly dependent, at least in part, on duration or severity. Furthermore, both systolic and diastolic function impairments were found to be independently associated with CS and hypertension. When considering treatment, partial improvement of structural alterations is expected after normalization of excess cortisol, whereas the possibility of improvements in functional disorders remains controversial. Although to what extent these findings are related to mortality of patients with overt CS or ACS remains vague, evidence presented in this review strongly demonstrates that hypercortisolemia itself has a relationship with geometric and/or functional alterations, as shown in clinical results as well as experimental findings. Nevertheless, the findings of this investigation require further elucidation regarding whether such alterations in pre-symptomatic state contribute to early diagnosis of hypercortisolemia, while evaluations of the effects of treatment on long-term mortality are also necessary.

## Figures and Tables

**Figure 1 jcm-11-07035-f001:**
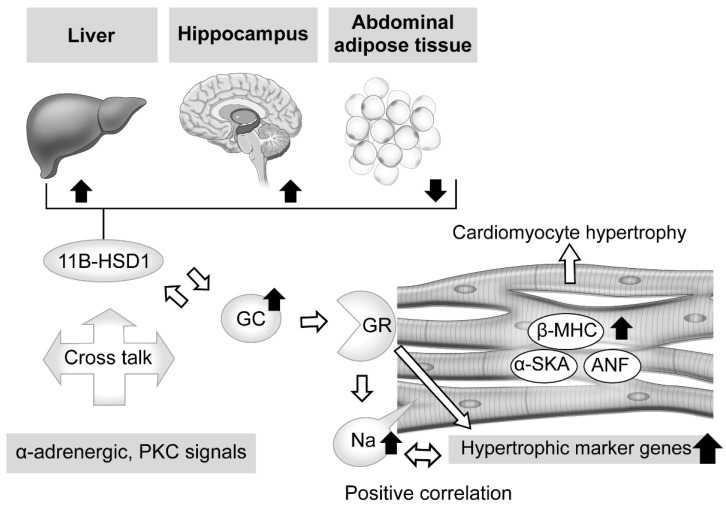
Effects of excessive glucocorticoid in cardiomyocytes and modulators of its signaling, including tissue-dependent regulation of 11β-hydroxysteroid dehydrogenase type 1. Excessive glucocorticoid (GC) enlarged the cell size of rat cardiomyocytes, with elevation of cardiomyocyte hypertrophic markers and related genes, such as atrial natriuretic factor (ANF), β-myosin heavy chain (β-MHC), and skeletal muscle α-actin (α-SKA). The glucocorticoid receptor (GR) antagonist and knockdown of GR expression attenuated that reaction, whereas suppression of mineralocorticoid receptor (MR) activity did not, indicating an essential role for GR in such a response. Phenylephrine (α-adrenergic receptor agonist) and phorbol ester (protein kinase C (PKC) activator), known to induce cardiomyocyte hypertrophy and related to hypertension and diabetes, potentiate the effects of corticosteroid and increase activated GR and MR expression. In addition, phorbol ester-induced hypertrophy exhibited increased 11β-hydroxysteroid dehydrogenase type 1 (11β-HSD1) expression and reductase activity. These results indicate possible cross talk between corticosteroid receptor-activated pathways, and both α-adrenergic and PKC signals. Corticosteroid also induces an increase in intracellular sodium concentration, which is related to ischemia, heart failure, and hypertrophy. That increase was revealed to have a positive correlation with hypertrophic gene expression. Furthermore, excessive corticosterone increased the expression level of 11β-HSD1 mRNA in abdominal adipose tissue, while that was decreased in the hippocampus and liver, suggesting regulation in a tissue-dependent manner.

**Figure 2 jcm-11-07035-f002:**
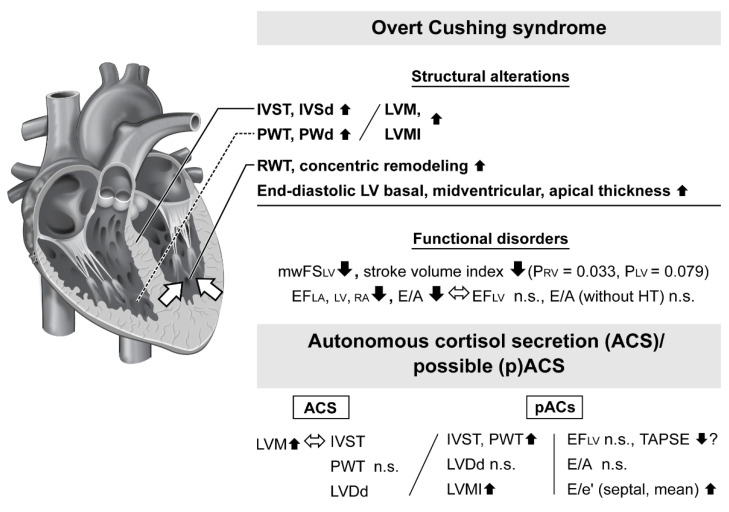
Structural and functional impairments related to overt Cushing syndrome (CS) and autonomous cortisol secretion (ACS), including possible (p)ACS. Overt CS patients show significantly greater interventricular septum thickness (IVST) and posterior wall thickness (PWT), as well as larger LV interventricular septum diameter (IVSd) and posterior wall diameter (PWd), with increased end-diastolic LV segmental thickness found in basal, midventricular, and apical slices. Furthermore, significantly thicker relative wall thickness (RWT) has been observed, and a hypertensive condition may promote increased left ventricular mass (LVM) and LVM index (LVMI) in overt CS patients. These structural alterations are associated with attenuation of heart performance, at least in regard to midwall fractional shortening (mwFS) and stroke volume index. As for ACA and pACS patients, only limited knowledge regarding structural and functional alterations is available. LVM or LVMI is significantly increased, while IVST, PWT, and LV diastolic diameter remain controversial. In patients with pACS, a slight reduction in tricuspid annular plane systolic excursion (TAPSE) with no impairment of LV ejection fraction (EF) may occur, as well as significantly higher values for septal and mean E/e’ without a decrease in E/A, suggesting that diastolic dysfunction above a certain level of might exist. LVM and LVMI were calculated as follows: LVM = 0.8 × [1.04 × (LVDd + IVST + PWT)^3^ − LVDd^3^] + 0.6; LVMI = LV mass/body surface area.

**Table 1 jcm-11-07035-t001:** Effects of hypercortisolemia on heart structure and function in Cushing syndrome patients.

Details and Numbers of Subjects	Age, Years	Sex, F/M	BMI, kg/m^2^	Duration to Diagnosis of CS (Months)	Complications (No.)	Assessment	Primary Findings	Reference
* Overt CS (12) *	27–57	8/4	-	-	Diabetes mellitus (7)	Echocardiography	LVH: (1) More severe IVST than essential hypertension and primary aldosteronism group (2) Significantly higher frequency of ASH Abnormalities in electrocardiogram: high-voltage QRS complexes and negative T wave in CS group Effects of treatment: improvement in echocardiography and normalization in electrocardiogram findings	[80]
CD (2)					Impaired glucose	Electrocardiogram		
Adrenal adenoma (9)					tolerance (4)			
Adrenal carcinoma (1)								
* CS (42) *	39 ± 12	30/12	28 ± 4.8	10.8 ± 14	Hypertension (32)	Echocardiography	LVH: larger IVSd and PWd, greater RWT and LVMI than those of the control Higher frequency of concentric remodeling in CS group Endocardial and mwFS: reduced in CS group LV diastolic function: reduction and prolongation in transmitral E and A flow velocities and E deceleration time in CS group	[81]
CD (36)					Diabetes mellitus (5)			
Adrenal adenoma (6)								
Adrenal carcinoma (1)								
Ectopic ACTH production (1)								
* CS (71) *	40.1 ± 1.46	61/10	29.4 ± 1.1	39.4 ± 5.49	Hypertensive	Echocardiography	LVH and remodeling: noted in up to 70% of patients LV systolic and diastolic functions: not impaired, including EF and E/A ratio LVH after remission: considerably ameliorated, but still more frequent Contributions to cardiac mass alteration: both excess cortisol and hypertensive condition	[82]
CD (65)					condition (30)			
Adrenal adenoma (2)								
Ectopic ACTH production (2)					Diabetes mellitus (8)			
ACTH-independent adrenal hyperplasia (2)								
* Overt CS (50) *	46.6 ± 14.3	44/6	–	70.8 ± 69.6	Hypertensive	Echocardiography	LVH: significant increase in LVMI and RWT, but not in IVST or PWT Geometric pattern: normal (35.1%), concentric hypertrophy (32.4%), concentric remodeling (24.4%), eccentric hypertrophy (8.1%) EFLV and E/A ratio: significant decrease in 24% and 62%, respectively, of patients with CS	[83]
Adrenal CS (50)					condition (41)			
					Diabetes mellitus (25)			
					Dyslipidemia (38)			
* CS (18) *	35.0	16/2	27.5	–	Hypertension (9)	Cardiac magnetic	Baseline cardiac parameters: lower EF in both sides of ventricle and LA, and increased end-diastolic LV segmental thickness as compared with control Effect of treatments: improvements in ventricular and atrial systolic performance, and decreased LVM Late gadolinium enhancement of myocardium in CS patients: absent in all	[84]
CD (15)	[26.2; 47.2]		[22.2; 33.0]		Diabetes mellitus (5)	resonance imaging		
Adrenal adenoma (2)					Impaired glucose tolerance (4)			
Ectopic ACTH production (1)					Dyslipidemia (6)			
* CS (25) *	44.6 ± 11.3	21/4	27.1	–	Hypertensive	Echocardiography	LVM and RWT: increased in patients with CS as compared with control Prevalence of pressure non-dipping profile: greater in CS patients than control, with no significant association with LVM or geometry Association between cortisol levels and LVM and RWT: no association between 24 h urinary cortisol levels and both LVM and RWT	[85]
CD (21)	(25.4; 32.4)				condition (16)			
Adrenal adenoma (4)					Diabetes mellitus (6)			
* Incidentaloma (46) *	69.3 ± 12.5	24/22	30.3 ± 7.5	–	Mean blood pressure (mmHg); 134 ± 9/82 ± 4	Echocardiography	Epicardial fat thickness: higher in both NFA and mild CS patients as compared with control LVM: highest for the CS group, and higher for NFA group than control Best correlation parameter with LVM in multiple regression analysis: epicardial fat thickness	[86]
Mild CS (6)								
NFA (40)					Mean fasting glucose (mg/dL): 128 ± 25.8			
* Incidentaloma (71) *	67	24/47	26.3	–	Hypertension (22)	Echocardiography,	LVMI: increased in patients with pACS as compared to NFA and mildly correlated with post-DST cortisol level LV functional impairments in pACS patients: higher prevalence of diastolic dysfunction than that of NFA Arterial stiffness in pACS patients: worse than that of NFA	[87]
pACS (34)	[59; 72]		[23.5; 29.4]		Diabetes mellitus (7)	brachial oscillometric		
NFA (37)					Dyslipidemia (24)	blood pressure waves		

Shown is a summary of findings of previous studies that evaluated heart structure and function in patients with Cushing syndrome (CS). A common structural alteration found was left ventricular hypertrophy (LVH), while relative wall thickness (RWT) and concentric remodeling were also concerning factors noted. Although systolic and diastolic function disorders remain controversial, midwall fractional shortening (mwFS), and E or A wave parameters may be useful. Interestingly, correlations of serum or urinary levels of cortisol with heart impairments were not routinely noted. Nevertheless, the diverse effects of cortisol should be considered. Effective treatment may improve these impairments, though long-term outcomes have not been fully elucidated. CS: Cushing syndrome; CD: Cushing disease; pACS: possible autonomous cortisol secretion; NFA: non-functioning adenoma; LV: left ventricular; LVH: left ventricular hypertrophy; LVM: left ventricular mass; LVMI: left ventricular mass index; IVST: interventricular septum thickness; IVSd: interventricular septum diameter; PWT: posterior wall thickness; PWd: posterior wall diameter; ASH: asymmetric septal hypertrophy; RWT: relative wall thickness; LA: left atrium; EF: ejection fraction; mwFS: midwall fractional shortening; DST: dexamethasone test. Data are expressed as range from minimum to maximum, mean ± SD, or median and interquartile range.

## Data Availability

Not applicable.

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
