# Peer review of "Cardiac Hypertrophy and Related Dysfunctions in Cushing Syndrome Patients—Literature Review"

_jcm, 2022, doi:10.3390/jcm11237035_

Round 1

Reviewer 1 Report

Reviewer comments on manuscript #jcm-2037439, entitled “Cardiac hypertrophy and related dysfunctions in Cushing syn-drome patients—need for early diagnosis and management

Comment n# 1, abstract section: “The survival rate of adrenal Cushing syndrome patients has been greatly reduced because of the availability of appropriate surgical and pharmacological treatments.” This phrase does not make sense – did the authors want to wrote - The survival rate of adrenal Cushing syndrome patients has been greatly increased?

Comment n# 2, main body section: the authors may also found interesting to explore the cardiac effects of GC in cases of apparent mineralocorticoid excess syndrome (OMIM: 218030), a rare autosomal recessive disorder caused by the presence of a severe deficiency of 11β-hydroxysteroid dehydrogenase type 2 (11βHSD2) activity, mainly due to multiple pathogenic variants in the HSD11B2 gene.

Author Response

Thank you for the good comments, which were helpful to improve the study. Please see our replies following.

Comment #1. We have corrected that sentence, as follows: The survival rate of adrenal Cushing syndrome patients has been greatly increased.

Comment #2. Thank you for the important suggestions. We are interested in apparent mineralocorticoid excess syndrome and, in fact, had presented additional information regarding 11βHSD2. However, we considered that further discussion was needed regarding the relationship between cortisol and aldosterone. Moreover, information regarding cardiac alterations, such as myocardial fibrosis due to mineralocorticoid excess, would also be needed. When considering the main point of this study and the desire to provide an appropriate article that the readers could easily understand, we have reluctantly eliminated discussion regarding that topic from the present manuscript.

Reviewer 2 Report

The authors have performed an excellent short revision of the current knowledge of the effect of hypercortisolism on cardiac function and structure, covering physiological aspects to clinical ones. Even though there is a lot of information on the cardiac comorbidity of Cushing syndrome, there has yet to be a recent comprehensive revision. It has been challenging to gather so much information and reach it thoroughly relating to basic and clinical aspects. Despite its complexity and the significant number of acronyms,  the article can be read without difficulties, giving integrated information on the effects of cortisol on the heart. However, even though it is evident, the paper has not been structured to answer the title's second part: the need for early diagnosis and treatment. Therefore this part of the title should be changed  

Author Response

Thank you for this essential suggestion. As the reviewer pointed out, the contents of the article did not satisfy the second half of the title, thus that part has been deleted. We trust that this change will be acceptable.